# A Sensitive Public Health Issue—The Vaccine Acceptancy and the Anti-Pertussis Immune Status of Pregnant Women from a Romanian Metropolitan Area

**DOI:** 10.3390/children10040640

**Published:** 2023-03-29

**Authors:** Valeria Herdea, Petruta Tarciuc, Raluca Ghionaru, Bogdan Pana, Sergiu Chirila, Andreea Varga, Cristina Oana Mărginean, Smaranda Diaconescu, Eugene Leibovitz

**Affiliations:** 1Pharmacy, Science, and Technology of Targu Mures, George Emil Palade University of Medicine, 540142 Targu Mures, Romania; 2Romanian Association for Pediatric Education in Family Medicine, 021507 Bucharest, Romania; 3Public Health and Management Department, Carol Davila University of Medicine and Pharmacy, 020021 Bucharest, Romania; 4Faculty of Medicine, Ovidius University of Constanta, 900470 Constanta, Romania; 5Medical-Surgical Department, Faculty of Medicine, Titu Maiorescu University of Medicine and Pharmacy, 031593 Bucharest, Romania; 6The Pediatric Infectious Disease Unit, Soroka University Medical Center, Beer-Sheva 85025, Israel; 7Faculty of Health Sciences, Ben-Gurion University of the Negev, Beer-Sheva 85025, Israel

**Keywords:** pertussis immunization, Tdap, pregnancy, pregnant women, early preventive health education

## Abstract

(1) Background: Immunization of pregnant women (PWs) against Bordetella pertussis infection is still a challenging health matter. (2) Methods: We gathered questionnaire data from 180 PWs regarding their expectancies and current opinion on infectious disease prevention. For the group of PWs who agreed to further investigations, the serum levels of Ig G anti-B. pertussis antibodies (IgG-PT) titer were measured and analyzed. (3) Results: A total of 180 PWs completed the questionnaire and 98 (54.44%, study group) accepted to perform the laboratory tests. During the first two pregnancy trimesters, PWs were found to be more willing (compared with the control group) to test for identifying high-risk situations that could affect themselves and their future infant (*p* < 0.001). Most of the participating PWs (91, 91.9%) had low levels of anti-pertussis antibodies (values < 40 IU/mL). Declared vaccine coverage of the PWs newborn infants for DTaP-1 and Prevenar 13 (at 2 months) and DTaP-2 and Prevenar 13 (at 4 months) vaccination reached 100% in the study group, while in the control group only 30/82 (36.59%) PWs accepted to be vaccinated during pregnancy, none of them providing data on their infants’ vaccine coverage. (4) Conclusions: Enrolled PWs faced a waning immunity against the B. pertussis infection. By raising maternal confidence in the protective role of vaccines against infectious diseases, better vaccine acceptance and better infant vaccine coverage can be achieved.

## 1. Introduction

Whooping cough (pertussis), a bacterial disease caused by Bordetella pertussis, may lead to severe respiratory disease in unvaccinated infants under 3 months of age [1,2,3,4,5,6].

Pertussis disease is underreported worldwide due to multiple reasons, including lack of data on disease incidence, regional differences, case definition differences used at the country level, reporting procedures or access to laboratory testing. In terms of the global burden of pertussis, Yeung et al. estimated 24.1 million pertussis cases globally and 160,700 deaths in children under 5 years of age in 2014 [7]. The disease can affect any age group, adolescents and adults being the main sources of infection for infants. The risk of developing severe forms is increased in pregnant women, unvaccinated young infants or immunosuppressed people. Pertussis antibodies transferred from the vaccinated mother to the newborn may decrease the risk of B. pertussis infection in the first months of life and protect the infant until reaching vaccination age [3,4,5,6,8,9,10,11].

The World Health Organization (WHO) initiated a global vaccination programme in 1974 for pertussis prevention. The transition from the whole-cell vaccine to the acellular pertussis vaccine reduced the potential side effects of the vaccines and subsequently improved the vaccine acceptance among various populations [12,13,14].

In Romania, the acellular pertussis vaccine was introduced in the National Vaccination Programme starting in the year 2011 for the completion of primary immunization (at the ages of 2, 4 and 11 months); first as part of a pentavalent vaccine and then, starting from 2013, as part of a hexavalent vaccine. The last dose of the pertussis vaccine needed to be administered at the age of six years. From 2018, 14 years-old teenagers may receive the combined vaccine with diphtheria, acellular pertussis, tetanus and poliomyelitis [15,16,17,18].

The last two decades revealed the vaccine hesitancy phenomenon, with a direct effect of a risky decrease in vaccine coverage all over the world. In 2018, more than 151,000 pertussis cases were globally reported by WHO. In 2021, during the SARS-CoV-2 pandemic, 25 million children under the age of 1 year did not receive basic vaccines; 18.2 million infants did not receive an initial dose of DTP and 6.8 million children were only partially vaccinated. One child in 1000 cases still dies from severe forms of whooping cough [19,20,21,22].

The incidence of pertussis among pregnant women reached 7.7/100,000 population in the United States during 2012–2017 [23] and 8.2/100,000 population in Europe between 2014–2018. Twenty-eight European countries have already recommended the anti-pertussis vaccination for pregnant women [24]. According to the Annual Epidemiological Report of the European Center for Disease Control, eight countries of the European Union decided in 2018 to introduce in their National Immunization Programmes the immunization against maternal tetanus, diphtheria and acellular pertussis (Tdap) for pregnant women [25]. More than 80% of infants with confirmed pertussis in the United States during 2009–2014 were younger than 2 months of age and needed hospitalization, and 81 infant deaths attributed to pertussis were recorded in the United States during 2009–2014. Of these, 61 (75%) were recorded in infants younger than 3 months of age [26,27].

The incidence of whooping cough in infants in Europe was reported as 44.4/100,000 [28]. By comparison, Romania reported 93 confirmed cases in 2018, 110 in 2019, 18 in 2020 and only one case in 2021 [29].

Starting with the first decade of the 20th century, boosting maternal antibody levels was explored with the active pertussis vaccination of pregnant women during the third trimester. No adverse maternal or neonatal events were documented, and elevated antibody titers were seen in both the mother and the infant [30,31]. Therefore, raising awareness of pertussis infection, improving the knowledge on the prevention of the disease in both adults and infants and analyzing the pertussis vaccine acceptance issues among Romanian PWs are of utmost importance.

The aims of the present study were (1) to develop and analyze the efficacy of a medical education programme on pertussis/whooping cough and its prevention among a cohort of PWs from Bucharest, Romania, during 2020–2022, and (2) to determine the immunity protective antibodies levels against pertussis among the enrolled cohort of PWs.

## 2. Materials and Methods

A prospective cohort study was developed during 2020–2022 in Bucharest, a metropolitan south-east area of Romania. In March 2020, a pilot webinar, with 18 PWs participants, allowed the evaluation and understanding of the participants’ expectations regarding the medical education programme development, their worries regarding vaccination safety during pregnancy, quality of their communication with the medical teams (family physicians teams and a gynecologist team for monitoring pregnancy) involved in their care and additional preferred themes to be discussed over the educational programme. The webinars were structured in simple, clear language and adapted to participants’ needs and general medical knowledge. According to the main goals of the study, and considering the identified needs following the pilot webinar, the final education programme was structured on three main themes: 1. infectious diseases prevention; 2. breastfeeding; 3. neurological development of the infant with a focus on infectious diseases prevention.

A total of 346 PWs aged 18–45 years were invited to attend an online, free-of-charge, informational medical education programme (Protect Life Ro) related to the prevention of infectious diseases with a focus on whooping cough. All participants were invited to complete a questionnaire before the programme initiation. The completion of the questionnaire was optional. The following steps included quantitative research of IgG anti-B. pertussis antibodies (IgG-PT) titers in the enrolled PWs and an evaluation of the vaccine acceptance decision-shaping factors (concerning their own and their infant’s vaccination), which was accompanied by a follow-up after these decisions (Figure 1).

Informed consent was obtained from all participants. The study’s first stage (questionnaire) benefits from the approval of the Ethics Committee of the Romanian Association for Pediatric Education in Family Medicine (AREPMF)—approval no 3/19/2020, from 19 March 2020. The study’s second stage (laboratory testing) benefits from the approval of the Academy of Medical Sciences—National Bioethics Commission of Medicines and Medical Devices under the number 2S/4, from 16 August 2021.

### 2.1. Study Phases

#### 2.1.1. Medical Education Programme (Protect Life Ro)

During 2020–2022, the researchers developed an education programme (Protect Life Ro) dedicated to PWs’ medical education. The programme allowed easy access to medical information on the prevention of various infectious diseases; it was adapted to the period of the SARS-CoV-2 pandemic (online access). Protect Life Ro facilitated participants’ access to webinars, medical websites, E-Books and various online information materials. This programme was hosted by VH (herdea.ro website), allowing future parents free access to medical information. The information regarding the education webinar programme was advertised by the researchers on social media networks before the established date of each webinar. Before participating in the medical education webinars, participants filled out an online questionnaire. A unique code was used for easy access to the questionnaire from any mobile phone or tablet. 

With the questionnaire, the participants received information about pertussis infection and an informed consent form. The questionnaire included 12 questions on various topics: Socio-demographic data (age, residence: rural/urban area and education).Pregnancy follow-up (gestational age, conception and number of offspring).PWs access to educational programmes (area of interest to access medical education, type of previous formal education, attitude of future mothers regarding vaccination acceptance and agreement for supply of educational materials).Vaccination acceptance was defined as the PWs willingness to be vaccinated (herself and/or her newborn baby) in general or specifically against whooping cough.

The questionnaire completion was mandatory for enrollment in stage 2 of the study (laboratory examinations).

#### 2.1.2. Evaluation of Pregnant Women’s Immune Status 

The IgG-PT antibody titers were determined using the ELISA method for testing pregnant women’s sera. Antibody detections were performed according to the guidelines of the European Reference Laboratory Center (INSTAND quality assessment, Germany; Lab Quality, Helsinki, Finland) [32,33].

The EUROIMMUN anti-B. pertussis toxin (PT) ELISA (IgG) is based on native highly purified pertussis toxin, which is produced only by B. pertussis. The test is quantitative, and results are expressed in international units per milliliter (IU/mL). The test is based on the first international standard of the WHO (WHO International Standard Pertussis Antiserum, human; 1st IS NIBSC Code 06/140) [34].

Recommendations on the evaluation of antibody titers against B. pertussis toxin (PT) of class IgG were as follows:Anti-PT IgG > 100 IU/mL: indication of an acute infection with B. pertussis (less than a year) or recent vaccination.Anti-PT IgG ≥ 50 IU/mL indicates an infection in the past few years.Anti-PT IgG < 40 IU/mL: no indication of an acute infection with B. pertussis, low-level antibodies.

The specificity with respect to the reference method was 95.5% with a sensitivity of 100% (excluding borderline results) [32,33].

#### 2.1.3. Effectiveness of the Educational Programmes during Pregnancy in the Vaccination Decision-Making Process

During 2020–2022, the PWs enrolled in the educational programme could access a minimum of three educational interventions: webinar, open access to the herdea.ro website, E-Books [35], newsletters with information on the Romanian National Vaccination Programme and infectious diseases up-to-date and reminders regarding the infant vaccination schedule from the Romanian National Vaccination Programme (e-mail messages).

Following delivery, the research team continued monitoring the study mothers and their infants. Parents from the study group received monthly short counselling messages (via e-mail or mobile phone) for one year regarding infant primary care network importance, breastfeeding, infectious diseases risk for the unvaccinated infant, the National Immunization Programme and basic infant home care. All the parents from the study group were asked to inform the investigators via e-mail or mobile phone messages about their infants’ immunization implementation during the first year of life, according to the National Immunization Programme. The control group received only one message, after birth, regarding the importance of the infant primary care network, infectious diseases risk for the unvaccinated infant and details on the National Immunization Programme.

### 2.2. Statistical Analysis

The statistical analysis was conducted using IBM SPSS Statistics version 26. We used univariate statistics to describe the mean, median and standard deviation for numerical variables (PW’s age, gestational age and IgG-PT values) or proportions for nominal values for the whole sample of 180 PWs and for each group (study group—98 PWs; control group—82 PW).

For testing the association in the case of nominal variables we used the Chi-square test. For comparing numerical continuous data, we used the T-test when comparing two groups, or ANOVA when comparing more than two groups, with post-hoc analysis if necessary. For determining the correlation levels between continuous variables, we used Pearson correlation. Binary logistics were used to determine the influence of variables on the PW’s decision to vaccinate.

We considered data to be normally distributed based on the aspect of the histogram and the results of the Shapiro–Wilk test. We considered that the results were statistically significant if a *p*-value less than 0.05 was obtained.

## 3. Results

Our initial online survey was carried out on 180 pregnant women (PWs). According to their acceptance to participate in the part of the study including the measurement of their IgG-PT values, participants were assigned to one of the two groups: the study group (PWs who agreed to perform the laboratory tests) and the control group (PW who did not agree to perform the laboratory tests).

### 3.1. Socio-Demographic Data

Socio-demographic data of the PWs is described in Table 1 and Figure 2a,b. The average age of PWs enrolled was 33.1 ± 4.8 years. Almost 16% of the women within the control group were above 40 years old, while in the study group the percentage was lower (8.2%). 

PWs in the control group had a higher gestation age at the initiation of the study (average 30.6 ± 5.2 weeks) compared with PWs belonging to the study group (26.9 ± 7.1 weeks)—*p* < 0.001. When analyzing the gestational age distribution (Figure 2b), in the study group the distribution was bimodal with approximately 10% of the PWs having a gestational age < 16 weeks. For the control group, almost half of the women were within the range of 32–36 weeks of pregnancy.

The overall percentage of enrolled PWs lived in urban areas (170, 94.4%). 

No statistically significant difference was observed in the educational level of the participants. Most of the PWs had studied in university (around three quarters of the participants in each group). There were more PWs who only completed elementary school in the control group (8.2%) compared to the study group (1.2%). 

Natural conception occurred in 82.7% of the PWs enrolled in the group and in 68.3% of the women belonging to the control group. In vitro fertilization (IVF) was reported for 10.2% of the pregnancies within the study group and in 22% of the pregnancies within the control group. Other means of conception were reported among 6% of the pregnancies within the study group and 3% of the pregnancies among the control group. No statistically significant difference was observed between the two groups with respect to the ways of conception (*p* = 0.105). When excluding the patients who did not provide a reply to the question concerning the way of conception, the percentage of pregnancies obtained by using artificial methods was significantly higher in the control group (29.1%. 23 pregnancies out of 79) compared with the study group (13.8%, 13 pregnancies out of 94)—*p* = 0.01.

For the enrolled PWs, we evaluated the degree of acceptance regarding their vaccination against pertussis before their participation in the Protect Life Ro series of webinars or before access to the educational materials developed in the project was available. The PWs within the control group (which also refused to evaluate their IgG-PT antibodies) disagreed on the need to be vaccinated against the disease in a higher percentage (39%) compared with the PWs from the study group (5%)-*p* < 0.001.

Around one-third of the PWs previously participated in parenting educational programmes (31.6% for the study group and 36.6% for the control group). Around one-quarter of the PWs showed no previous interest in participating in educational programmes dedicated to pregnant women; without statistically significant differences between the two groups (*p* = 0.605) (Figure 3).

From all options offered by the educational programme (online webinar, E-Books, printed materials and classroom), the online educational programme model was preferred by most of the participants (86.7%). A secondary option was represented by E-Books (70%) (Figure 4). A significant difference was observed between the study group and the control group with respect to the preference for E-Books as a means of getting the information (54.1%-study group, 89%-control group, *p* < 0.001).

### 3.2. Laboratory Examinations

Most of the participating PWs (91, 92.9% of the study group) had low levels of anti-pertussis antibodies. The results were inconclusive in 4.1% of the PWs (suggesting potential disease or vaccination in recent years), while the examinations were positive for 3.1% of the PWs, indicating a recent infection or vaccination.

The average value of IgG-PT for the 91 PWs with negative results was 8.4 IU/mL ± 6.7 IU/mL (median 5.16 IU/mL).

The correlation level between the age of the PWs and their antibody levels was a very weak, negative one, r = −0.058, and not statistically significant (*p* = 0.588). A linear correlation between the age of the PWs and the levels of IgG-PT could not be identified (Figure 5). 

### 3.3. Impact of the Educational Programme on the Personal Anti-Pertussis Vaccination Uptake 

We analyzed the PWs attitude towards personal vaccination during pregnancy by evaluating their opinion before the initiation of the educational programme and completion of vaccination during pregnancy (Table 2). All PWs (from both the study and control group) who agreed with vaccination from the beginning received the pertussis vaccine during pregnancy (except for those for whom the results of the IgG-PT indicated inconclusive or protective levels, making the vaccination unnecessary). For the study group, all PWs who initially did not agree to be vaccinated, and also those who mostly agreed to be vaccinated, decided to receive the vaccine during pregnancy. For the control group, the results indicate that there was no change in in the attitude versus vaccination for the PWs who did not agree with the vaccination, while 33% of the ones who mostly agreed initially decided to receive the vaccine during pregnancy.

Regarding the overall infant vaccination, no data was available for patients within the control group. For PWs within the study group, 80.6% of their infants had completed the national vaccination scheme at the age of 4 months.

We ran a binary logistic analysis to determine the role of various included variables (decision to evaluate IgG-PT level, age of PW, gestational age, educational level, type of conception and previous participation in parenting classes) in the final decision of the PWs to vaccinate against pertussis during pregnancy. The logistic regression model was statistically significant: χ^2^(8) = 43.606, *p* < 0.001. The model explained 30.1% (Nagelkerke R2) of the variance in the decision to vaccinate and correctly classified 69.9% of the cases. The area under the ROC was 0.784 (95% CI 0.716–0.851), which is an acceptable level of discrimination according to Hosmer et al. (2013) [36]. Of the six predictor variables, three were statistically significant: PWs age, gestational age and previous participation in parenting classes (Table 3).

## 4. Discussion

The main findings reported in this study bring about a focus on a sensitive public health issue, namely the topic of pregnant women’s immunization with the purpose to protect the newborn. There is a reluctance to vaccinate pregnant women in many countries, which is unjustified from the point of view of evidence-based medicine [1,2,3,4,5,6,7,8,9,10,11,12,13,14,23,24,25,26,27,28,29,30,37]. On the other hand, in the last two decades, the phenomenon of vaccine hesitancy spread considerably in the general population [19,20,21]. Nevertheless, pregnant women are considered to be a group at risk for the occurrence of infectious diseases [23,24,26]. In Romania, many factors contributed to a lower vaccine acceptance for this category, represented mainly by the frequent changes in the children and teenagers’ immunization schedules and types of vaccines. Other relevant factors included inconstant supplies for many vaccines [31], lack of a National Immunization Programme for adult immunization, the 2020–2022 COVID pandemic (which raised vaccine hesitancy at the population level and also among pregnant women), access to misleading information, the influence of fake news, frequent negative media campaigns within the last decades and various myths regarding PWs immunization [11,38]. As a result, waning immunity in childbearing-aged women could be observed.

One of our goals was to encourage and raise the level of PWs’ medical education and particularly to raise awareness of the whooping cough disease risk. PWs were invited to access the Protect Life Ro educational programme during 2020–2022, and 346 participated in the programme; out of which 180 filled out an online survey. The questionnaire analysis revealed several “red flags alerts” regarding the factors that can influence the acceptance of vaccination in PWs.

The average age of the PWs participating in the study was 33 years. Around three quarters of the participants in each group had graduated from university, meaning that PWs who have access to a high educational level were indeed open to following an educational programme during pregnancy. In Romania, in 2019, the average mother’s age at first birth was 29 years in urban areas and 25 years in rural areas, while the average age at which a mother gives birth to her first child in the EU was 29.4 in 2019 [39,40,41]. It is known that developed countries are struggling with similar issues regarding the access to medical care in rural areas, starting with limited resources of healthcare providers and finishing with limited access to financial resources, compared with the urban areas [42,43].

Many factors were described as able to shape the PWs attitude regarding immunization during pregnancy: lack of knowledge regarding the risk of infectious diseases, lack of a national awareness programme for PWs, lack of adequate counselling from the physicians’ teams who take care of the pregnancy, fear of a vaccines’ composition, fake news or vaccine safety issues. International data show variable uptake rates (between 0–78%) for anti-pertussis vaccination in PWs [19,20,21,44,45,46].

Interestingly, we reported in this study that a significantly higher proportion (29.1%) of PWs that used artificial methods of conception belonged to the control group, compared with the study group (13.8%); a finding that deserves confirmation and analysis in additional studies.

The second part of the study focused on evaluating the IgG-PT levels in PWs in the study group.

We showed that 92.9% of the study group had unprotective anti-pertussis antibody levels; 3.06% had a protective level of antibodies suggesting recent clinical pertussis infection and 4.08% of them had low antibody titers (above 50 IU/mL, suggesting clinical pertussis infection in the last 3–5 years) [47].

The protective titer of IgG-PT antibodies is maintained 4–7 years post-vaccination or 12–20 years after the disease [11]. According to the Romanian National Vaccination Programme, anti-pertussis vaccination must be initiated with the primary vaccination schedule at 2, 4 and 11 months, and followed by boosters at 6 and 14 years of age. Although a booster is recommended throughout adult life, no adult anti-pertussis booster immunization has been carried out in Romania during the last 10 years [15,16,18,37].

An unprotective value of IgG anti-PT (below 40 UI/mL) exposes the PWs to a high risk of contracting whooping cough and, of course, being unable to transmit protective antibodies to the new-born, leading to a lack of protection against pertussis of the newborn during the first months of life until vaccination [48].

When evaluating the efficacy of the educational programme provided to the PWs, we focused our research on two aspects: implementation of vaccination during pregnancy and vaccination of their infants during the first year of life.

Despite much controversy around the subject of PWs immunization, the large majority of the medical staff counsels PWs in a pro-immunization manner. We must admit that a small part of healthcare professionals, maybe not involved directly in the immunization process, may show a hesitant attitude regarding PWs vaccination [45,46,49]. Not all specialties’ curricula include immunization classes. As a future solution, immunization programmes could be part of the continuous medical education for all medical specialties.

The vaccination coverage for pertussis in PWs reached variable rates on the European continent. In Spain, a study that analyzed 36,032 clinical records registries of PWs from primary care centers in respect to the acellular pertussis vaccine use showed coverage rates varying between 49.8% in 2016 and 79.4% in 2018 [49]. In the UK, the mean coverage rate for pertussis vaccine coverage in PWs during 2021–2022 was 64% [50]. There is no official data on pertussis vaccination coverage for PWs in Romania. In the present study, following participation in the educational programme, 92.9% of the PWs accepted to be vaccinated against whooping cough (seven did not have an indication for vaccination according to their antipertussis IgG levels). Furthermore, even for the participants in the control group, we noticed a significant change in acceptance of antipertussis vaccination, as one-third of the initially reluctant PWs eventually decided to receive the vaccine during pregnancy.

For our secondary goal, the infant vaccination for 0–12 months after birth, we compared our data with the most recent estimates for children born in 2020 from the Romanian National Center for Study and Infectious Diseases Control [51]. The report indicates a 76.4% vaccine coverage at the age of 12 months of the diphtheria-tetanus-acellular pertussis vaccine (three doses, administered at 2, 4, 11 months).

The data of the WHO regarding the DTP-containing vaccines global coverage for 2021 show 86% (first dose, two months), 80% (second dose, four months) and 81% (third dose, eleven months) [52].

For the PWs enrolled in the study group, 80.6% of their infants had completed their vaccination scheme at the age of 4 months, and these findings are definitely encouraging. Although the study groups were small, our findings may serve as an interim guide on educating further mothers to reduce hesitancy towards maternal vaccination within the societal environment from which the study population was recruited, with possible extrapolation to other populations.

It is important to mention that a severe decline in vaccine coverage was registered for all types of vaccines for PWs and infants during the COVID-19 pandemic, mainly related to the pandemic restrictive rules, inconstant vaccine supplies, fear of parents regarding COVID-19 disease and fake news regarding vaccines side effects [45,53].

We also obtained encouraging results in the control group, where the PWs refused to perform the laboratory tests but requested to access the educational programme. In this group, we obtained feedback reporting on 30 PWs vaccinations against pertussis, with one-third of them changing their initial anti-vaccination opinion, showing that the educational programme applied in early pregnancy could raise vaccine confidence and facilitate the decision-making process in the prevention of pertussis.

Our study has some limitations related to the lack of a bigger cohort study, collection of data only from a geographical metropolitan area of Romania, lack of statistical data from Romania regarding PWs antipertussis vaccination, enrolment in the study of predominant “high skills” educated population and the emergence of the COVID-19 pandemic period that overlapped with the progress of the study.

## 5. Conclusions

Our study showed that PWs are facing a concerning waning immunity against the B. pertussis infection.

We also showed that awareness gaps are present among PWs regarding the role of infectious disease prevention by vaccination. Health education in pregnancy can act as a shaping factor for the informed decision of future mothers, reducing the vaccination hesitancy phenomenon and raising vaccine confidence regarding prevention against B. pertussis infection. Empowering childbearing-aged women with knowledge can facilitate the decision for the immunization of their future infant against severe forms of infectious disease.

## Figures and Tables

**Figure 1 children-10-00640-f001:**
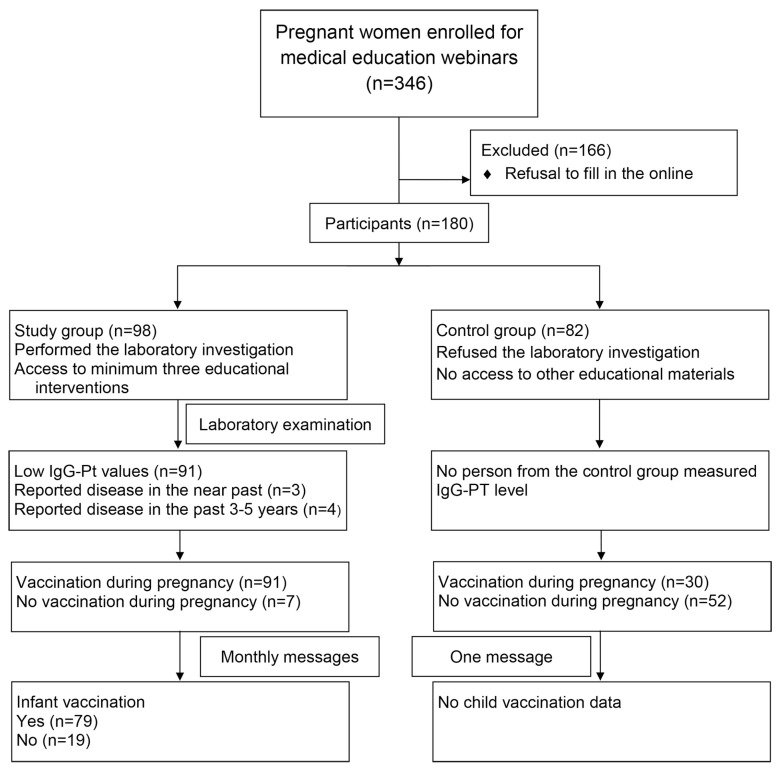
Study design (flow sheet).

**Figure 2 children-10-00640-f002:**
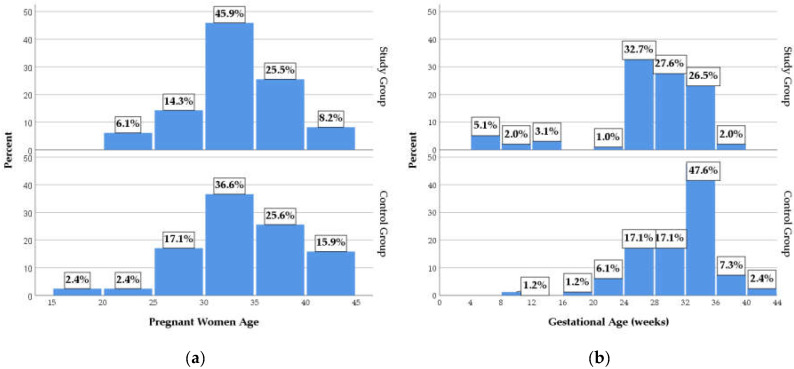
(**a**) Distribution according to the age of the PWs; (**b**) distribution according to gestational age.

**Figure 3 children-10-00640-f003:**
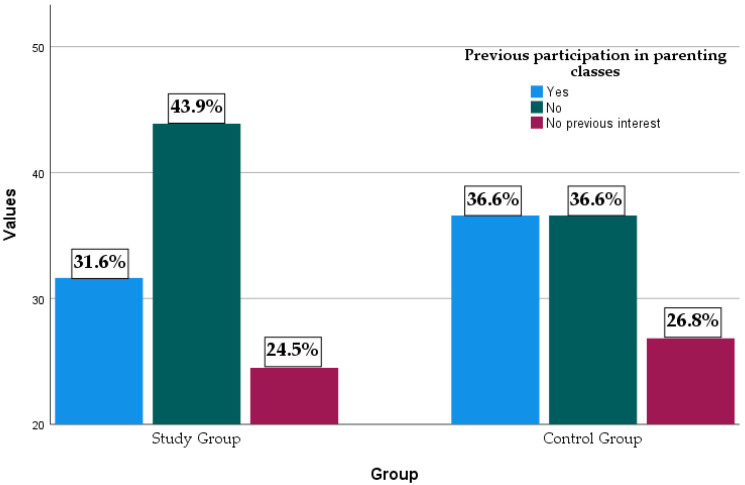
Distribution of enrolled PWs according to previous participation in parenting classes.

**Figure 4 children-10-00640-f004:**
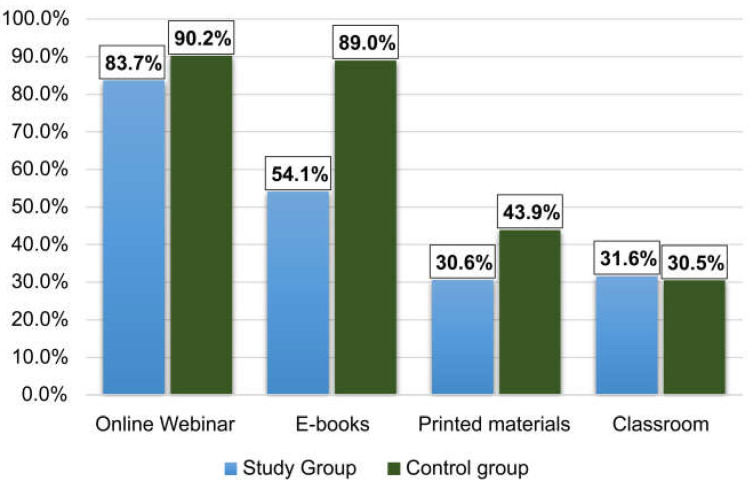
Educational programme model preferences.

**Figure 5 children-10-00640-f005:**
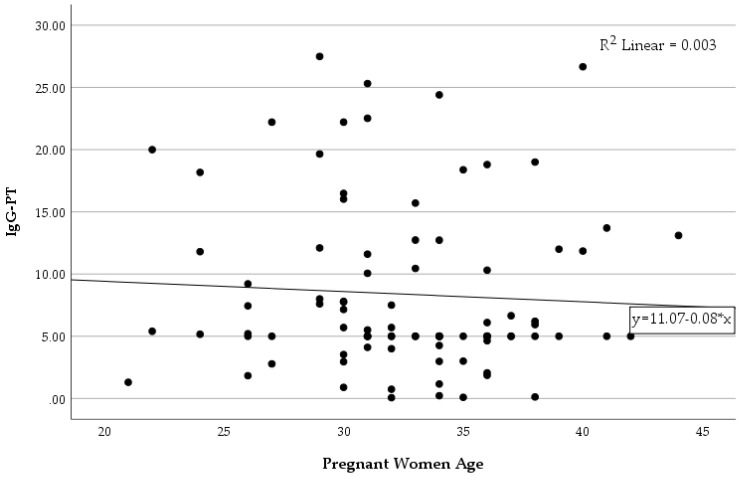
Regression PW age-IgG-PT levels (in patients with a non-protective level of antibodies).

**Table 1 children-10-00640-t001:** Socio-demographic data (180 enrolled pregnant women).

	Total (n = 180)	Study Group * (n = 98)	Control Group ** (n = 82)	*p* Value
Age (years)				
Mean ± SD	33.1 ± 4.8	32.7 ± 4.7	33.5 ± 4.9	0.23
Median	33	32.5	34	
Gestational age (weeks)				
Mean ± SD	28.6 ± 6.6	26.9 ± 7.1	30.6 ± 5.2	<0.01
Median	29	28	32	
Environment				
Urban	170 (94.4%)	88 (89.8%)	82 (100%)	<0.01
Rural	10 (5.6%)	10 (10.2%)	0 (0%)	
Educational Level				
Elementary School	9 (5%)	8 (8.2%)	1 (1.2%)	0.08
High School	40 (22.2%)	19 (19.4%)	21 (25.6%)	
University	131 (72.8%)	71 (72.4%)	60 (73.2%)	
Type of Conception				
Natural Conception	137 (76.1%)	81 (82.7%)	56 (68.3%)	0.11
In Vitro Fertilization (IVF)	28 (15.6%)	10 (10.2%)	18 (22%)	
Other Means	8 (4.4%)	3 (3.1%)	5 (6.1%)	
No Answer	7 (3.9%)	4 (4.1%)	3 (3.7%)	
Vaccination acceptance				
Agree	80 (44.4%)	60 (61.2%)	20 (24.4%)	<0.01
Mostly agree	63 (35%)	33 (33.7%)	30 (36.6%)	
Disagree	37 (20.6%)	5 (5.1%)	32 (39.0%)	

* Completed the Protect Life Ro informational programme + completed the questionnaire + performed the serologic tests; ** did not complete the Protect Life Ro informational programme + completed the questionnaire + performed the serologic tests.

**Table 2 children-10-00640-t002:** Impact of the educational programme on the personal anti-pertussis vaccination uptake: comparison between study and control groups.

Group			Attitude towards Personal Vaccination during Pregnancy						Total	
			Agree		Mostly Agree		Don’t Agree			
			N	%	N	%	N	%	N	%
Control Group	Personal vaccination	No	0	0.0%	20	66.7%	32	100.0%	52	63.4%
		Yes	20	100.0%	10	33.3%	0	0.0%	30	36.6%
Study Group	Personal vaccination	No	7 *	11.7%	0	0.0%	0	0.0%	7	7.1%
		Yes	53	88.3%	33	100.0%	5	100.0%	91	92.9%

* Did not vaccinate because lack of recommendation (according to antibody titers results).

**Table 3 children-10-00640-t003:** Logistic regression predicting likelihood of vaccination in pregnant women based on age, gestational age, educational level, type of conception and previous participation in parenting classes.

	B	S.E.	Wald	df	*p*	Odds Ratio	95% C.I. for Odds Ratio	
							Lower	Upper
Pregnant Women Age (years)	−0.093	0.042	4.839	1	0.028	0.911	0.838	0.990
Gestational Age (weeks)	−0.175	0.047	14.121	1	0.000	0.839	0.766	0.920
Educational Level (Elementary)			0.005	2	0.997			
Educational Level High School)	21.092	15,076.986	0.000	1	0.999	1.45 × 10^9^	0.000	
Educational Level (University)	0.032	0.429	0.005	1	0.941	1.032	0.445	2.392
Type of conception (Natural)			1.086	2	0.581			
Type of conception (FIV)	0.809	0.809	0.999	1	0.317	0.445	0.091	2.176
Type of conception (Other)	0.594	0.891	0.444	1	0.505	0.552	0.096	3.166
Previous participation in parenting classes (Yes)			7.596	2	0.022			
Previous participation in parenting classes (No)	0.706	0.450	2.457	1	0.117	2.026	0.838	4.899
Previous participation in parenting classes (No Previous Interest)	1.295	0.470	7.589	1	0.006	3.650	1.453	9.170
Constant	8.971	2.339	14.705	1	0.000	7873.210		

## Data Availability

The datasets used and/or analyzed during the current study are available from the corresponding author upon reasonable request.

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
