# Peer review of "A Sensitive Public Health Issue—The Vaccine Acceptancy and the Anti-Pertussis Immune Status of Pregnant Women from a Romanian Metropolitan Area"

_children, 2023, doi:10.3390/children10040640_

Round 1

Reviewer 1 Report

This is an insightful manuscript that highlights the degree to which vaccine hesitancy has spread through communities facilitated by internet -enabled devices. Unfortunately, this wave of hesitancy is impacting not only potential mothers but also newborn infants (<2 months) who are dependent on pregnant mothers to make vaccine choices. The manuscript describes the efforts to educate pregnant mothers to be vaccinated against pertussis, an infectious disease which disproportionately targets newborn infants. This is an important topic that warrants publication. As the manuscript acknowledges, though the study groups are small the results are encouraging, and the study could serve as interim guide on educating to-be mothers to reduce hesitancy towards maternal vaccination within the societal environment from which the  study population was recruited. Hwever, prior to bein published the manuscript requires changes some changes to more clearly describe its objectives. It is also advised that it undergo some close editing to catch typographic errors and sentences that are quite unwieldy to understand in one reading. I recommend major changes to provide sufficient time to restructure the presentation of the study.

Comments  

Introduction

The Introduction is somewhat confusing and needs to be more focussed in outlining the main problem the study is trying to address, and in relating the study to the title of the manuscript. It unnecessarily introduces a large section on general pertussis vaccination that is only partially relevant while distracting from the objectives that need to be highlighted. I would suggest restructuring/rewriting parts of the Introduction using the last paragraph to guide the Introduction, stressing on the background of the problem and the relevance of the study:  how interventional education programs could improve pertussis vaccination rates for pregnant women in a vaccine-hesitant environment .

Materials and Methods Section 2.1

Was the pilot medical program conducted earlier on the 18 PWs  to structure the main study ? This is not clear. If it was, this should be mentioned earlier in section 2.1 and not at the end.

Results:

FIG 2: Percentage numbers are a bit small to read comfortably and should be larger.

They should be consistently presented in all the graphs (either all on top of the bars as in Fig 2b or as insets within the bar as in Fig 2a)

Fig 5: The rationale for why correlations between IgG titers and Age was analyzed and presented (without any correlations detected)

The source of information that titers of  >40 IU anti-pertussis IgG titers  are protective should also be cited in the main manuscript and not only in the Materials and Methods. Citing any more recent mechanistic study would be helpful (example: Healy et aL Association Between Third-Trimester Tdap Immunization and Neonatal Pertussis Antibody Concentration. JAMA. 2018; 320(14):1464-1470. PMID: 30304426)

Out of interest were the IgG titers and its relevance of the titers communicated to the participants ?

The structure of the Results section would be clearer for the reader if the different Socio-demographic categories of the study groups is included under one subsection as (3.1). Followed by the section on Laboratory analysis  as (3.2), and then  Impact of the educational program as (3.2)

Discussion

The manuscript concludes that having access to a high level of education signifies more openness to following an education program. I am confused on how this conclusion was drawn. The education levels between the control and study group were similar, so how does education and openness correlate based on your data?

It is interesting that all the rural participants (n=10) participated in the study group. Does this warrant a sentence in the discussion ?

Example of minor edits needed:

LN: 189 Control group received “one message” not “once message”

Ln 209 “consisting of”  not “consisting in”

Example of disconnected contradictory sentences needing rephrasing (LN224-225)

“The overall group of enrolled PWs lives mostly in urban areas (170, 94.4%). All participants in the study living in rural areas were part of the study group.”

Author Response

March 20, 2023

Dear Editor:

Thank you for considering for publication our manuscript titled “A sensitive public health issue - The vaccine acceptancy and the anti-pertussis immune status of pregnant women from a Romanian metropolitan area”. We have corrected/modified the manuscript in accordance with the reviewers’ suggestions (the corrections are marked in red) as follows:

Reviewer no.1

Comments  

Introduction

The Introduction is somewhat confusing and needs to be more focussed in outlining the main problem the study is trying to address, and in relating the study to the title of the manuscript. It unnecessarily introduces a large section on general pertussis vaccination that is only partially relevant while distracting from the objectives that need to be highlighted. I would suggest restructuring/rewriting parts of the Introduction using the last paragraph to guide the Introduction, stressing on the background of the problem and the relevance of the study:  how interventional education programs could improve pertussis vaccination rates for pregnant women in a vaccine-hesitant environment .

The introduction was considerably shortened and restructered.

Materials and Methods Section 2.1

Was the pilot medical program conducted earlier on the 18 PWs  to structure the main study ? This is not clear. If it was, this should be mentioned earlier in section 2.1 and not at the end.

The section on the pilot medical program conducted earlier on 18 PWs was moved at the beginning of the materials and method section.

Results:

FIG 2: Percentage numbers are a bit small to read comfortably and should be larger.

They should be consistently presented in all the graphs (either all on top of the bars as in Fig 2b or as insets within the bar as in Fig 2a).

All figures were redesigned for consistency and for better visualisation.

Fig 5: The rationale for why correlations between IgG titers and Age was analyzed and presented (without any correlations detected).

We think that this figure provides important data and enriches the presentation of the results.

The source of information that titers of >40 IU anti-pertussis IgG titers  are protective should also be cited in the main manuscript and not only in the Materials and Methods. Citing any more recent mechanistic study would be helpful (example: Healy et aL Association Between Third-Trimester Tdap Immunization and Neonatal Pertussis Antibody Concentration. JAMA. 2018; 320(14):1464-1470. PMID: 30304426)

We added relevant citation for clarifying the interpretation of the IgG titers and the cut-off points used in the discussions section.

Out of interest were the IgG titers and its relevance of the titers communicated to the participants ?

Yes, this information was communicated to participants.

The structure of the Results section would be clearer for the reader if the different Socio-demographic categories of the study groups is included under one subsection as (3.1). Followed by the section on Laboratory analysis  as (3.2), and then  Impact of the educational program as (3.2)

We divided the results section in 3 subsections, as suggested.

Discussion

The manuscript concludes that having access to a high level of education signifies more openness to following an education program. I am confused on how this conclusion was drawn. The education levels between the control and study group were similar, so how does education and openness correlate based on your data?

The reviewer is right, we deleted this conclusion, as we did not find any differences in the education levels of the two groups.

It is interesting that all the rural participants (n=10) participated in the study group. Does this warrant a sentence in the discussion ?

We think that the data on the very few rural participants was not indicative and conclusive, and deleted it.

Example of minor edits needed:

LN: 189 Control group received “one message” not “once message”

Corrected.

Ln 209 “consisting of”  not “consisting in”

Corrected.

Example of disconnected contradictory sentences needing rephrasing (LN224-225)

“The overall group of enrolled PWs lives mostly in urban areas (170, 94.4%). All participants in the study living in rural areas were part of the study group.”

Deleted, we agree it was not relevant.

Reviewer 2 Report

Dear Editor,

Thank you for the kind invitation to review this manuscript. Overall, the manuscript is well written.

Abstract

- What does O immunisation mean?

Introduction

- The authors did a great job in performing literature review to highlight the problem on hand regarding pertussis vaccination

-> However, it will help to highlight medical education programs performed and efforts done in Romania to improve vaccination rates 

-> The introduction while comprehensive may benefit from some degree of summarisation

Methods

- it will be helpful to provide the questionnaire and to comment if the designed questionnaire was piloted in any of the population

- How was the decision to focus on the 3 themes in line 155 made?

- How was vaccine acceptance defined?

Results

- Suggest to report age to 1 decimal place

- Table 1 suggest to provide categorical groups for the gestational age

- Figure 2 and 3 are too small and not legible

- In table 2, there is a category of "mostly agree" - what does this entail?

Discussion

- Are there any unique findings of this study?

- How does this study compare to other studies for similar educational program

- What are the implications of the study findings and room for improvement?

- Given the recent covid-19 pandemic which noted high rates of vaccine hesitancy, a research area to look into perhaps is to see if there are correlation of vaccine hesitancy broadly or specific to certain vaccines.

-> Can cite this: https://pubmed.ncbi.nlm.nih.gov/34452026/

Author Response

Reviewer no. 2

Abstract

What does O immunisation mean?

Deleted.

Introduction

The authors did a great job in performing literature review to highlight the problem on hand regarding pertussis vaccination

However, it will help to highlight medical education programs performed and efforts done in Romania to improve vaccination rates 

The introduction while comprehensive may benefit from some degree of summarisation

The introduction was shortened considerbaly and restructured.

Methods

it will be helpful to provide the questionnaire and to comment if the designed questionnaire was piloted in any of the population

We consider that our manuscript provided in the methods sections all the details presented in the questionnaire; the manuscript is already extremely extended and we think that adding all the questionnaire details will not add any important information. We rearanged the provided information on the questionnaire for clarity.

How was the decision to focus on the 3 themes in line 155 made?

As mentioned in the text, the decision to focus on the 3 themes was based on the reply of the 18 pregnant mothers from the pilot study.

How was vaccine acceptance defined?

We added in the methods section a sentence defining the concept of vaccine acceptance.

Results

Suggest to report age to 1 decimal place

We would like to present the data as 2 decimals, as generally accepted.

Table 1 suggest to provide categorical groups for the gestational age

We did not have these data. Available.

Figure 2 and 3 are too small and not legible

All figures were redesigned for consistency and for better visualisation.

In table 2, there is a category of "mostly agree" - what does this entail?

„Mostly agree”: as understandable, the categoty is of those mothers who are not fully convinced of the vaccination advantages but also do not reject it; some half way between acceptance and rejection

Discussion

Are there any unique findings of this study?

We emphasized the unique findings of our study in the discussion and conclusion  section; Our study showed that PWs are facing a concerning waning immunity against the B. pertussis infection and also showed that health education in pregnancy can act as a shaping factor for the informed decision of future mothers.

How does this study compare to other studies for similar educational program

We are not aware of similar studies, combining an educational program for pregnant women with a laboratory investigation of their immunization status and with an analysis of further steps (following the completion of the educational program) on their personal vaccination and the vaccination of their infants

What are the implications of the study findings and room for improvement?

We mentioned in the discussion section the limitations of our study: a relative small study group, performed in only one specific area of the country and some enrollment biases, all these parameters requiring a larger future study (on whooping cough and also on other diseases common in newborns and very young infants, like influenza, COVID-19 and particularly RSV).

Given the recent covid-19 pandemic which noted high rates of vaccine hesitancy, a research area to look into perhaps is to see if there are correlation of vaccine hesitancy broadly or specific to certain vaccines.

We agree, but it was not part of the objectives of our study.

The Editor:

We completed extensive English revisions of the manuscript.

Prof. Eugene Leibovitz

Round 2

Reviewer 1 Report

Thank you for making the revisions and addressing all my points raised.

The amendment to the structure of the Result section has inadvertently been overlooked (See below). Please ensure that this is made in the proofs of the manuscript.

> The structure of the Results section would be clearer for the reader if the different Socio-demographic categories of the study groups is included under one subsection as (3.1). Followed by the section on Laboratory analysis  as (3.2), and then  Impact of the educational program as (3.2) 

We divided the results section in 3 subsections, as suggested.

Author Response

Dear Editor:

Thank you for considering for publication our manuscript titled “A sensitive public health issue - The vaccine acceptancy and the anti-pertussis immune status of pregnant women from a Romanian metropolitan area”. We have re-corrected/modified the manuscript in accordance with last reviewers’ suggestions (the corrections are marked in red) as follows:

Reviewer no.1

Thank you for making the revisions and addressing all my points raised.

The amendment to the structure of the Result section has inadvertently been overlooked (See below). Please ensure that this is made in the proofs of the manuscript.

 > The structure of the Results section would be clearer for the reader if the different Socio-demographic categories of the study groups is included under one subsection as (3.1). Followed by the section on Laboratory analysis as (3.2), and then Impact of the educational program as (3.2) 

Thank you for this observation. We added the missing section 3.1 Socio-demographic data and renumbered the other two subsections of the result accordingly.

Prof. Eugene Leibovitz

Reviewer 2 Report

The authors commented that the age should be reported to 2 decimal places

- Kindly report as 1 decimal place as the 2nd decimal place does not add any value to the results. to also standardize the number of significance places for the results

The font in figure 1 is too small and require editing to make them legible. 

Otherwise nil further comments

Author Response

Dear Editor:

Thank you for considering for publication our manuscript titled “A sensitive public health issue - The vaccine acceptancy and the anti-pertussis immune status of pregnant women from a Romanian metropolitan area”. We have re-corrected/modified the manuscript in accordance with last reviewers’ suggestions (the corrections are marked in red) as follows:

Reviewer no. 2

The authors commented that the age should be reported to 2 decimal places.

- Kindly report as 1 decimal place as the 2nd decimal place does not add any value to the results. to also standardize the number of significance places for the results

We made the changes in the manuscript and tables and report results with one decimal place. We also reduced the number of decimals for all significance values.

The font in figure 1 is too small and require editing to make them legible. 

We re-designed figure 1 for better legibility

For the editor: we received for second revision an uncorrected manuscript (where our initial track changes were not incorporated, probably because your decision was to correct all the track changes after receiving also our second revision). Therefore, we made the new modifications (addressed in this reply letter) by new track changes in this manuscript, in addition to the initial track changes. In addition, we made additional minor changes to the English styling of our manuscript.

Prof. Eugene Leibovitz
